# Real-Time Multi-Class Disturbance Detection for Φ-OTDR Based on YOLO Algorithm

**DOI:** 10.3390/s22051994

**Published:** 2022-03-03

**Authors:** Weijie Xu, Feihong Yu, Shuaiqi Liu, Dongrui Xiao, Jie Hu, Fang Zhao, Weihao Lin, Guoqing Wang, Xingliang Shen, Weizhi Wang, Feng Wang, Huanhuan Liu, Perry Ping Shum, Liyang Shao

**Affiliations:** 1Department of Electrical and Electronic Engineering, Southern University of Science and Technology, Shenzhen 518055, China; 11930535@mail.sustech.edu.cn (W.X.); fhyu21@foxmail.com (F.Y.); 11853004@mail.sustech.edu.cn (S.L.); 11849550@mail.sustech.edu.cn (D.X.); 12031313@mail.sustech.edu.cn (J.H.); 12031197@mail.sustech.edu.cn (F.Z.); 11510630@mail.sustech.edu.cn (W.L.); 20034303r@connect.polyu.hk (X.S.); liuhh@sustech.edu.cn (H.L.); shenp@sustech.edu.cn (P.P.S.); 2Department of Electrical and Computer Engineering, Faculty of Science and Technology, University of Macau, Macau 999078, China; 3Department of Microelectronics, Shenzhen Institute of Information Technology, Shenzhen 518172, China; wanggq@sziit.edu.cn; 4The Department of Electronic and Information Engineering, Hong Kong Polytechnic University, Kowloon, Hong Kong, China; 5Peng Cheng Laboratory, Shenzhen 518005, China; wangwzh@pcl.ac.cn; 6College of Engineering and Applied Sciences, Nanjing University, Nanjing 210023, China; wangfeng@nju.edu.cn

**Keywords:** distributed fiber sensing, Φ-OTDR, real-time detection, multi-class classification, object detection, YOLO

## Abstract

This paper proposes a real-time multi-class disturbance detection algorithm based on YOLO for distributed fiber vibration sensing. The algorithm achieves real-time detection of event location and classification on external intrusions sensed by distributed optical fiber sensing system (DOFS) based on phase-sensitive optical time-domain reflectometry (Φ-OTDR). We conducted data collection under perimeter security scenarios and acquired five types of events with a total of 5787 samples. The data is used as a spatial–temporal sensing image in the training of our proposed YOLO-based model (You Only Look Once-based method). Our scheme uses the Darknet53 network to simplify the traditional two-step object detection into a one-step process, using one network structure for both event localization and classification, thus improving the detection speed to achieve real-time operation. Compared with the traditional Fast-RCNN (Fast Region-CNN) and Faster-RCNN (Faster Region-CNN) algorithms, our scheme can achieve 22.83 frames per second (FPS) while maintaining high accuracy (96.14%), which is 44.90 times faster than Fast-RCNN and 3.79 times faster than Faster-RCNN. It achieves real-time operation for locating and classifying intrusion events with continuously recorded sensing data. Experimental results have demonstrated that this scheme provides a solution to real-time, multi-class external intrusion events detection and classification for the Φ-OTDR-based DOFS in practical applications.

## 1. Introduction

After being proposed in 2005 [1], the phase-sensitive optical time-domain reflection technique has been widely used [2] in geological exploration [3,4,5], partial discharge monitoring [6,7], traffic sensing [8,9], marine health monitoring [10], and perimeter security [11,12], etc. It allows long-distance (10 km or more) vibration monitoring on sensing fiber [13,14]. The DOFS system can locate the position of the disturbance in the spatial domain and acquire the vibration information of the disturbance in the temporal domain.

Analyzing vibration information and classifying it into different types is a research hot spot in this area. A lot of works are proposed based on traditional classification algorithms that use human-extracted signal features for learning to classify disturbances [15]. Wang et al. use the relevance vector machine (RVM) to learn the features extracted by wavelet analysis and achieved 88.6% on the classification problem [16]. Sun et al. artificially extracts multiple features and perform correlation analysis for dimensionality reduction on the disturbance signals in the spatial–temporal domain and use three RVM classifiers to classify the three types of intrusions, achieving an accuracy rate of 97.8% [17].

These traditional classification algorithms belong to “expert systems”, and they require human-determined features. However, these features will become meaningless in real complex engineering applications. For example, the laying method of fiber under test (FUT) and light source quality may bring uncontrollable factors. These uncontrollable factors will cause the correlation of the human-determined features to decrease, and the traditional methods will no longer be applicable. Therefore, a convolutional neural network (CNN) is required for automatic feature extraction and disturbances classifying in complex situations. Using CNN instead of human-determined features brings stronger robustness to real-world applications. Wu et al. used 1-D CNN to classify five types of events in pipeline monitoring and achieved 98% accuracy [18]. Wang et al. used a deep dual-path network (Deep DPN) to classify the disturbances, and 97% accuracy is obtained [12]. However, the deeper network structure not only brings longer training time and increases the training burden, but also fails at real-time operation in actual application scenarios: the computation time needed to classify the sensing signal is several times more than the acquisition time. Therefore, it is necessary to design an algorithm that can quickly and accurately locate and classify external disturbances to meet the real-time operation demand in practical scenarios.

To quickly locate and classify the sensing signal, there is currently a method combining DOFS and YOLO algorithm proposed by Zhou et al. in 2021 [19]. They took the lead in the application of YOLO in pipeline inspection gauge detection. YOLO is a fast object detection algorithm, which can determine the bounding box and classify events at the same time in a network [20]. However, the method they proposed can only detect a single type of event. Therefore, different from their work, we demonstrate a new method for detecting and classifying multi-class disturbance events and achieving real-time operation, making it more suitable for practical applications.

This paper proposes a real-time multi-class disturbance detection algorithm for Φ-OTDR sensing system based on YOLO algorithm, which can quickly and accurately locate, and classify multi-class disturbance. Firstly, to achieve fast detection speed, this method turns two-steps (traditional object detection: locate THEN classify) into one-step (one network), like its name, YOLO: “You Only Look Once”, which completes the same task with less computation complexity, so the detection speed can be improved compared with the traditional algorithms. Secondly, to achieve precise locating and multi-class classification, this method based on the advanced YOLO Network for training and testing. As a result, a real-time multi-class disturbance detection scheme for Φ-OTDR-based DOFS is provided to the community, which we believe will have a positive effect on practical applications, especially online monitoring scenarios.

## 2. Principle of Operation

The distributed optical fiber sensing system we use is the direct detection Φ-OTDR and the experimental setup is shown in Figure 1a. The pulsed light is driven into the sensing fiber, and the original one-dimensional sensing data is obtained through the Rayleigh backscattering (RBS) signal returned by the circulator. We arrange the RBS traces brought back by each pulse in sequence, convert into a two-dimensional spatial–temporal sensing data matrix, a typical data structure of DOFS, as shown in Figure 2, in which the horizontal axis represents the space domain, and the vertical axis represents the time domain. We calculate the differential between two adjacent RBS traces to demodulate the external vibration [1]. We normalize the differential result, save it as an image, and finally label the event on the image to complete the construction of the dataset. We use the open-source tool named labelImg from Github to label the intrusion events in the spatial–temporal sensing images, label the type and location of events and save them in a txt file. The images and the corresponding label information will be used for supervised learning.

The workflow of the YOLO-based real-time multi-class disturbance detection algorithm is shown in Figure 1b, which mainly includes five stages: signals acquisition, data preprocessing, data labeling to form a dataset, training the YOLO network, and using the well-trained model for testing. As the physical resolution of the sensing system is affected by the pulse width of high frequency pulsed light, the nearby data of the vibration center contains rich information. The data matrix near the center point as the sample of the disturbance signal for positioning and labeling, and the labeling method meets the requirements of the training set of the original YOLO algorithm. YOLO is pursuing the optimal speed and accuracy trade-off for real-time applications. As shown in Figure 3, the network has two main components, the first part uses Darknet53 for feature extraction, and the second part uses Feature Pyramid Networks (FPN) for feature fusion to generate the prediction results at three scales. Darknetconv2d_BN_Leaky (DBL) is the smallest component of Darknet53, which is used to do the two-dimensional convolution operations. DBL contains convolution (conv), batch normalization (BN) and nonlinear activation function (LeakyReLU). Resblock is the main component in Darknet53 and consists of DBL and n residual units. Residual unit refers to ResNet and solves the degradation problem caused by increasing the number of layers in the network [21]. YOLO uses FPN to generate three different scales of feature maps, which can be used for cross-scale prediction. Therefore, our YOLO-based scheme has great detection ability of tiny-sized objects with almost no reduction in detection speed, which is why it is suitable for localization and classification of weak disturbance events in long-range sensing information.

Randomly divide the dataset as train set (70%) and test set (30%). The training set is used to adjust the weight of the network, and the test set is used to verify the generated network model and focus on the accuracy and detection speed of the algorithm for the detection and classification of disturbance events.

It should be noted that before the YOLO network is trained with the Φ-OTDR dataset, the idea of transfer learning is adopted [22]. YOLO uses the ImageNet data for pre-training [23]. In CNN, different depths of convolutional layers have different functions for extracting image features. In image processing, most of the first few layers extract the common features of training data such as color blobs and Gabor filters, and subsequent layers are trained according to the requirements of the specific tasks. Therefore, ImageNet dataset is used for pre-training. The parameters of the first 20 layers from the pre-trained model are retained, and the remaining parameters are initialized randomly to form the initial model of our algorithm for Φ-OTDR dataset. Pre-training using a large dataset like ImageNet can effectively improve the network’s image processing capabilities and shorten the time required for training [22].

## 3. Experiment and Result

### 3.1. Distributed Optical Fiber Sensing System & Data Collection

The Φ-OTDR system used in this work is shown in Figure 1a. It uses a narrow-linewidth laser (NLL, 1550 nm) with 5 kHz linewidth and 23 mW output power as a light source and an acousto-optic modulator (AOM) to transform the continuous laser into pulsed light. The pulse width is 100 ns, and the repetition frequency rate is 60 kHz. We use the erbium-doped fiber amplifier (EDFA) to amplify the optical signal, which can compensate for insertion loss and transmission loss. The amplified pulsed light is driven into the sensing fiber through the circulator. As the light pulse advances, the RBS light carrying different position vibration information returns along with the fiber to the circulator and is output from the circulator 3 port to the second EDFA for re-amplification. At the end of the optical system, the spontaneous emission noise from the EDFA is filtered by a fiber Bragg grating (FBG). The RBS signal is finally fed into the photodetector (PD), completing the conversion from optical signals to electrical signals. Finally, the data acquisition (DAQ) device records the data with a sampling frequency of 240 MSa/s.

In this experiment, the optical fiber is laid on a metal protective net and the ground near the net, to detect 5 types of events, which are calm state (I), rigid collisions against the ground (II), hitting the protective net (III), shaking the protective net (IV), and cutting the protective net (V). These events act on different positions of the sensing fiber. In order to reduce the cost of data collection, we used the FUT (1.6 km) laying method shown in Figure 4. One vibration event is detected by multiple sections of the FUT and subsequently treated as multiple signal samples.

The specific number of samples collected for each event is shown in Table 1, and the schematic diagram of sensor images for different events is shown in Figure 5.

The details of the 5 events are as follows:(I)Calm state

Collect the signal under the ordinary environment, the main component is the environmental noise, no one interferes.

(II)Rigid collisions against the ground

In order not to damage the outdoor ground, a hammer (536 g) dropped from a height of 10 cm is used as a representative of rigid collision to collect data. The rigid collision formed the pattern in Figure 5II, which was in line with our expected result of the hammer falling.

(III)Hitting the protective net

We use a hammer to hit the upper and middle beams at different positions of the metal protective net (1.4 m × 1.45 m) at a stable frequency and use random forces to simulate the impact. The regular blue–red pattern appearing in Figure 5III is consistent with the characteristics of stable frequency. The patterns appearing at 1340 m and 1380 m are caused by the FUT laying method. It can be observed that the three groups of patterns at 1340 m, 1360 m and 1380 m do not affect each other.

(IV)Shaking the protective net

The experimenters faced the protective net, grasped the grid of the protective net, and shook it with normal strength and frequency. Our shaking causes the protective net to shake within a range of 15° from front to back, with a frequency between 1 Hz and 3 Hz. Such an event eventually formed a diagonally staggered blue–yellow pattern as shown in Figure 5IV, which was highly recognizable.

(V)Cutting the protective net

The optical fiber is laid in an S-shape on the protective net and is used to emulate the behavior of cutting the net. If the fiber is cut along with this behavior, it can be clearly found in the sensing information; if the protective net is cut off, but the optical fiber is not broken, the optical fiber will fall naturally. Use cable ties to fix the fiber on the protective net and cut the cable tie to simulate the natural fall of the second situation and the corresponding pattern is shown in Figure 5V.

To avoid that the disturbance classification is influenced by its occurrence location during disturbance detection (one FUT location corresponds to a specific disturbance type), it is necessary to decouple the disturbance type from its occurrence location. Therefore, after the data acquisition of the “cutting the protective net “(V) at each location, all the remaining ties are cut and the FUTs of other different areas are re-laid on the net and the board according to Figure 4.

### 3.2. Data Pre-Processing

After using a photodetector to convert the optical signal into an electrical signal, the DAQ is used for data collection. The one-dimensional sensing data is subsequently converted to spatial–temporal sensing matrix as shown in Figure 2, whose horizontal axis direction is the spatial domain, and the vertical axis direction is the time domain. The moving average method is adopted to suppress the random noise of the raw data [24]. We calculate the differential between two adjacent RBS traces and normalize the differential result. The processed 2D data will be stored as an image, and the amplitudes are converted into the color of each pixel. The converted images are labeled according to the type of vibration event within and stored for network training and testing.

### 3.3. Comparison between YOLO and Traditional Detection Algorithms

With the development of computer vision, more image object detection algorithms have been proposed, among which the most representative ones are RCNN and its improved versions Fast-RCNN and Faster-RCNN. RCNN was proposed by Ross Girshick in 2014 [25]. It is the pioneering work of object detection using deep learning. He innovatively combined Selective Search, CNN, and Support Vector Machine (SVM) to do object detection for images.

However, because of its CNN computation for all region proposals, the same feature extraction task was repeated many times. Therefore, there is a large time cost for both training and testing. Fast-RCNN uses ROI (region of interest), and use softmax to replace the SVM in RCNN, and uniformly maps the bounding box information to the feature map [26]. Compared with RCNN using stretching for normalization, ROI reduces the repeated calculation of the layer before feature extraction, thus speeding up the calculation. Faster-RCNN uses RPN (Region Proposal Network), which generates a bounding box faster to replace Selective Search, and completes an end-to-end CNN object detection model, which improves the overall operating speed [27]. The differences of structure between algorithms are shown in Figure 6.

These three algorithms are based on the two-stage scheme of traditional image object detection. First determine where it is (determine the bounding box), then determine what it is (classification). The YOLO object detection algorithm we use innovatively proposes one-stage, which uses a single network to complete the traditional two-step work, which further improves the speed and achieves real-time operation. Therefore, YOLO is widely used in autonomous driving and video surveillance. We use Fast-RCNN, Faster-RCNN and YOLO (all three schemes were pre-trained using ImageNet dataset) to compare their performance in spatial–temporal sensing images, and the results are shown in Figure 7.

Since one event will produce multiple continuous patterns on the spatial–temporal sensing images, and the detection results of different algorithms are quite different, we explained the indicators (correct locating and correct classification) and explained detecting results of the “calm state (I)”.

Locating

We believe that after an event occurs, if one of the multiple patterns caused by the event is recognized as any event, it can be considered as successful locating. If a pattern corresponding to no event is located as any event, it is a false alarm, not a misclassification.

We use the result of the perturbation detection to locate the location of the perturbation. More specifically, we use the horizontal coordinate of the center point of the bounding box to locate the perturbation. Therefore, the accuracy of localization is highly dependent on the quality of perturbation location labeling in the training set.

2.Classification

Only the classification results of the data considered as “events” will be counted, and the confusion matrix will be calculated and drawn.

For example, a pattern caused by a “shaking the protective net (IV)” is classified as “hits the protective net (III)”, we think this is a misclassification but successfully located. 

3.About (I) calm state

The detecting results of the “calm state” will be displayed only if no disturbance event is found. Therefore, when something happens, our special treatment of the “calm state” reduces the redundant information display and makes the disturbance event we are concerned about more visible.

We train these three algorithm models by GeForce GTX 1080 Ti with 12 GB memory and compare their performance on the same hardware. After being well-trained, the three algorithms have a good performance on the same dataset, all of which are 100% located to the event with no false alarm, and the classification accuracy rate has reached more than 95.737%. Although the accuracy of the YOLO-based algorithm is slightly inferior to the Faster-RCNN, it has a unique advantage in detection speed, as shown in Table 2. The speed is 44.90 times that of Fast-RCNN and 3.79 times that of Faster-RCNN. YOLO uses FPN for feature fusion, so it has better detection results for tiny objects in large data in principle. Therefore, we believe that the YOLO algorithm can be applied to detect the disturbance events on DOFS dataset.

### 3.4. Real-Time Sensing Video Processing

The faster computing speed achieved by YOLO meets the demand for real-time processing, so further experiments are carried out using brand new data which has not been included in the previous dataset. We use the same system parameters for continuous data collection. After slicing and pre-processing the raw data, a sensing signal up to 30 s is obtained. In order to be more suitable for industrial scenes, we converted the matrix into a video by applying a sliding window to the original signal matrix during data acquisition. A window length *T_l_* of 0.5 s is used for 30 s video generation, as shown in Figure 8, and the frame rate of the video is 20 FPS, corresponding to a sliding step *t_s_* of 0.05 s. From the previous experiments, it can be seen that the processing time *T_p_* for a sensing image with *T_l_* of 0.5 s is 0.0438 s. As *T_p_* is less than the sliding step *t_s_* between two adjacent frames, real-time operation is achieved with the proposed method.

If we consider the events required for other steps such as DAQ sampling, data pre-processing, etc., we can solve the real-time problem by increasing the sliding step *t_s_* and decreasing the frame rate of the video. As long as all other events *T_other_* plus *T_p_* are smaller than *t_s_*, the real-time operation still holds.

The sensing video contains the above-mentioned 5 types of events, and there are situations where multiple events occur at the same time. We used the well-trained model to detect disturbance events in the video, and the detection results are shown in Figure 9. At different moments in the video, each disturbance event was detected separately. In Figure 9a, the monitoring results of the “Calm State” are presented as no other disturbing events are detected, which is in accordance with our expectation and presentation logic. In Figure 9b, a “Rigid Collision” event is detected at 1019 m of the sensing fiber (position information is obtained by mapping the coordinates of the center point of the bounding box to the actual sensing distance). In Figure 9c, the behavior of the “Hit Net” is detected twice in sensing image, and the mean value of the centroids of the two bounding boxes corresponds to 1486.63 m, and the pattern corresponds to 1484.72 m, which have a small error (1.9 m). In Figure 9d, we fixed two sections of sensing fiber at 130 m intervals to the protective net, so that the “Shaking Net” behavior was detected at 1335.65 m and 1480.91 m, respectively. In Figure 9e, we cut the tie to allow the sensing fiber that was secured to the protective net to fall and collide with the net. At 1411.81 m, our method recognizes the “Cut Net” event and does not misidentify several other patterns. The collision of the fiber with the net in the “cut net” event also affects other fibers in proximity (laid according to Figure 4). In Figure 9f, two events have been detected at the same time (“Rigid Collision” at 681.95 m and “Hit Net” at 1106.72 m), demonstrating the multi-class vibration detection ability of the proposed method. The results present that there is no missed detection of vibration events or misclassification in real-time detecting of sensing video.

## 4. Conclusions

This paper proposes a real-time multi-class disturbance detection method based on YOLO algorithm for Φ-OTDR. We use CNN-based methods to automatically extract features, avoiding the low robustness problem of “expert systems” in complex environments. Using the YOLO algorithm based on Darknet53 and FPN, real-time monitoring can be performed on spatial–temporal sensing data acquired from the Φ-OTDR system. The spatial-temporal signal collected from the Φ-OTDR system is converted into images after pre-processing, and manually labeled according to the location and types of external disturbance as a dataset. In the experiments, it only costs 0.0438 s on average to complete the locating and classification of intrusion events for 0.5 s sensing data when treated as an image. Meanwhile, when the sensing data is converted to a video of 20 frames per second, it achieves real-time operation for locating and classifying intrusion events with continuously recorded sensing data. Experimental results prove that our proposed scheme has achieved the real-time operation (22.83 FPS, which is 44.90 times faster than the Fast-RCNN and 3.79 times faster than the Faster-RCNN) while ensuring high accuracy (96.14%) in five types of disturbance detection. The proposed method provides a promising solution for real-time multi-class disturbance detection for industrial application of Φ-OTDR, especially for online monitoring scenarios.

## Figures and Tables

**Figure 1 sensors-22-01994-f001:**
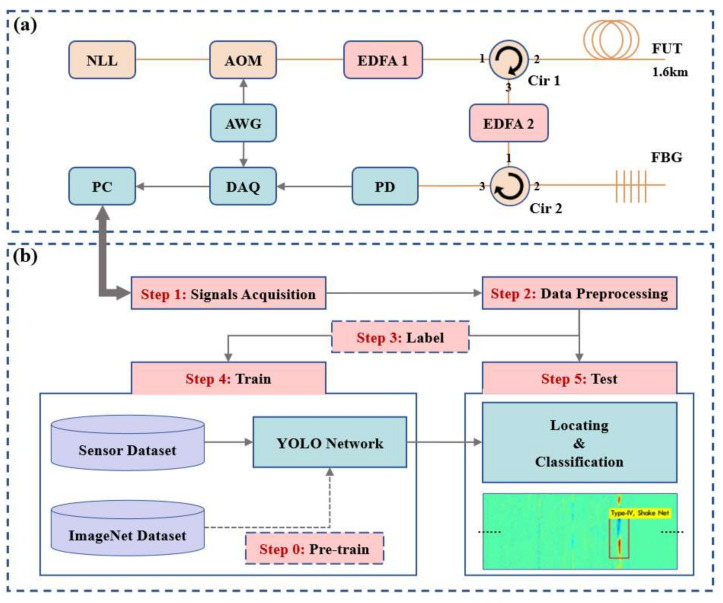
(**a**) Experimental setup of the direct detection Φ-OTDR. (**b**) Workflow of the real-time multi-class classification disturbance detection algorithm. NLL: narrow-linewidth laser; AOM: acousto-optic modulator; EDFA: erbium-doped fiber amplifier; Cir: circulator; FBG: fiber Bragg grating; AWG: arbitrary wave generator; PD: photodetector; DAQ: data acquisition card; PC: personal computer. As shown in step 5, the results of locating and classification are shown in the following image (partial magnification of the detection results).

**Figure 2 sensors-22-01994-f002:**
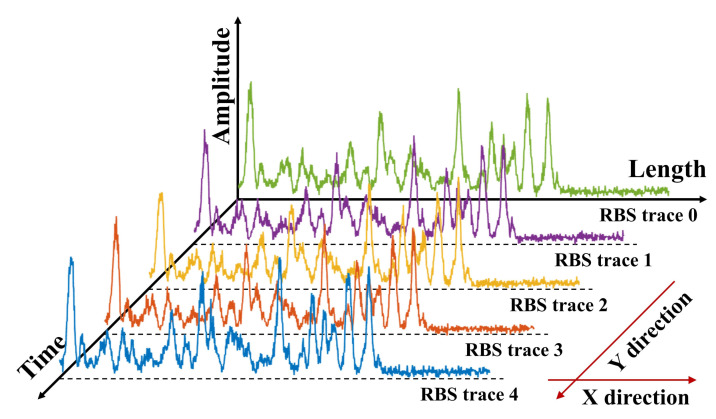
The schematic diagram of the spatial-temporal sensing matrix.

**Figure 3 sensors-22-01994-f003:**
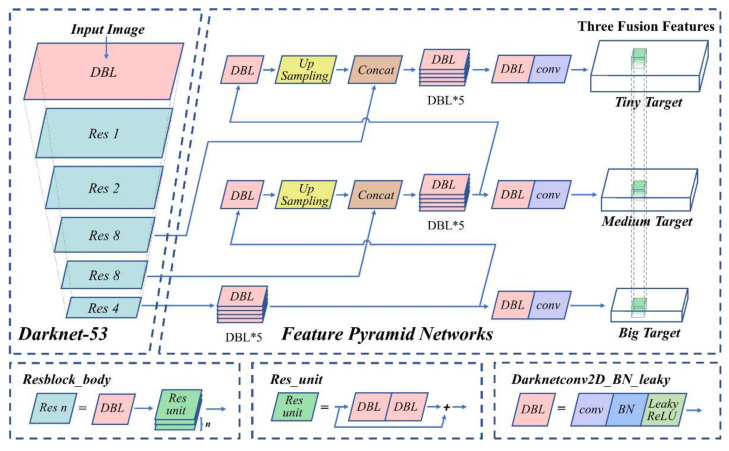
Network structure of YOLO-based real-time multi-class classification disturbance detection algorithm. DBL: Darknetconv2d_BN_Leaky; Res: Resblock_body; Res unit: residual unit; Up-Sampling: increase the dimensions of the image by interpolation; Concat: concatenates features for feature fusion; conv: convolution; BN: batch normalization; LeakyReLU: a type of nonlinear activation function.

**Figure 4 sensors-22-01994-f004:**
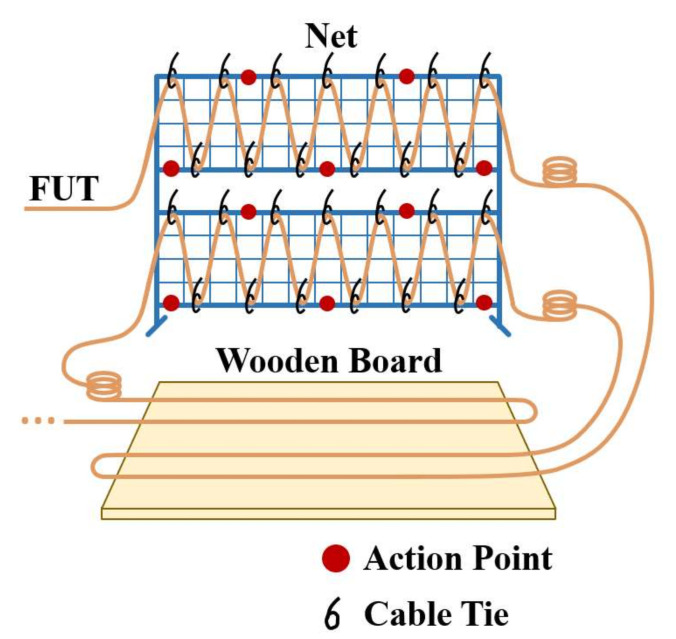
FUT laying method: multi-point sensing experiment on protective net and wooden board.

**Figure 5 sensors-22-01994-f005:**
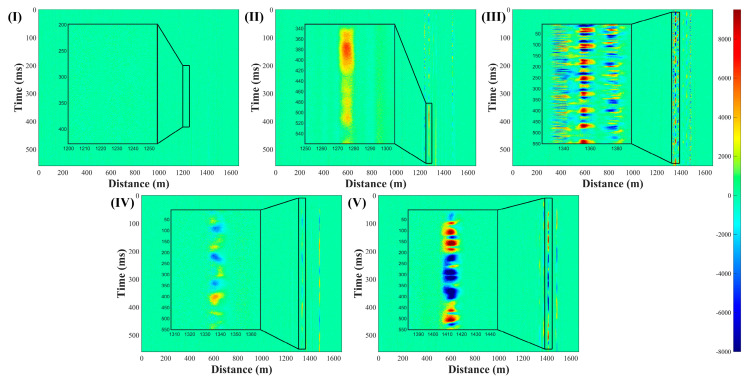
The spatial–temporal sensing image of 5 events: (**I**) calm state; (**II**) rigid collisions against the ground; (**III**) hitting the protective net; (**IV**) shaking the protective net; and (**V**) cutting the protective net. All black boxes are partial magnifications of the detection results.

**Figure 6 sensors-22-01994-f006:**
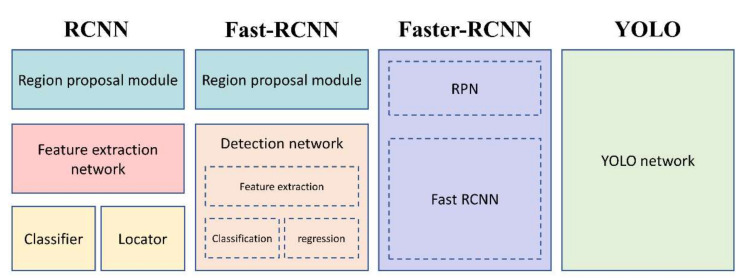
Schematic diagram of the workflow and structure of RCNN, Fast-RCNN, Faster-RCNN and YOLO.

**Figure 7 sensors-22-01994-f007:**
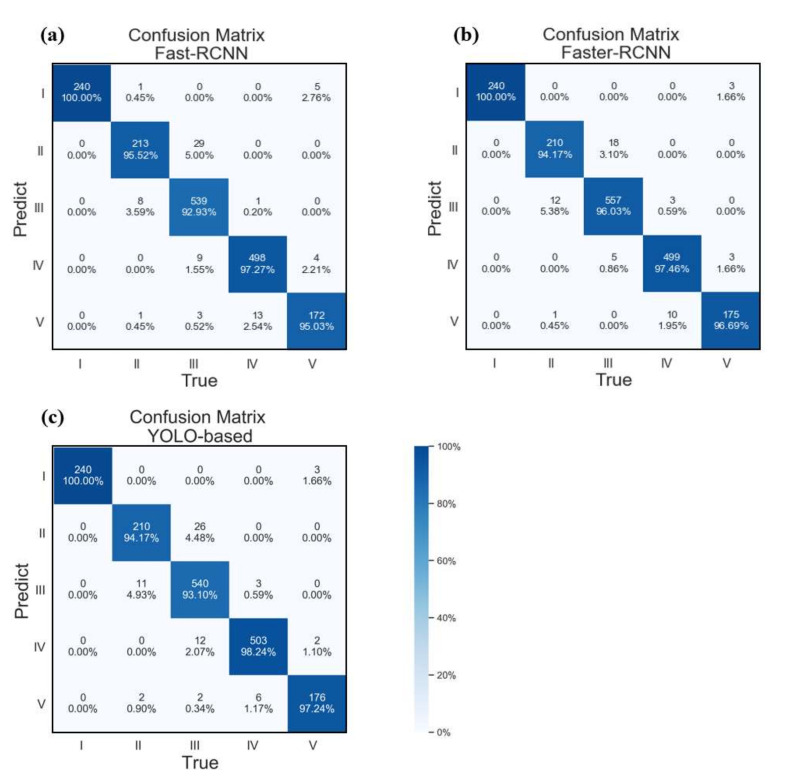
Confusion matrix of Fast-RCNN (**a**), Faster-RCNN (**b**) and YOLO-based scheme (**c**).

**Figure 8 sensors-22-01994-f008:**
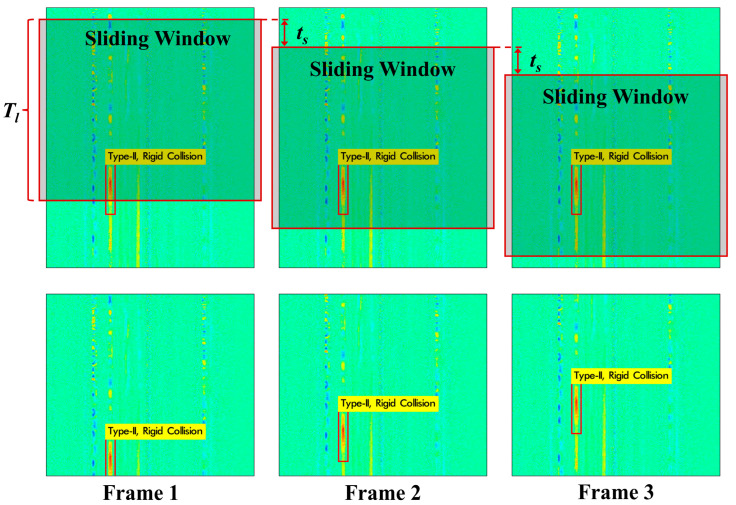
Schematic diagram of sensor image generation based on Sliding Window Principle. *T_l_*: sliding window length; *t_s_*: sliding step.

**Figure 9 sensors-22-01994-f009:**
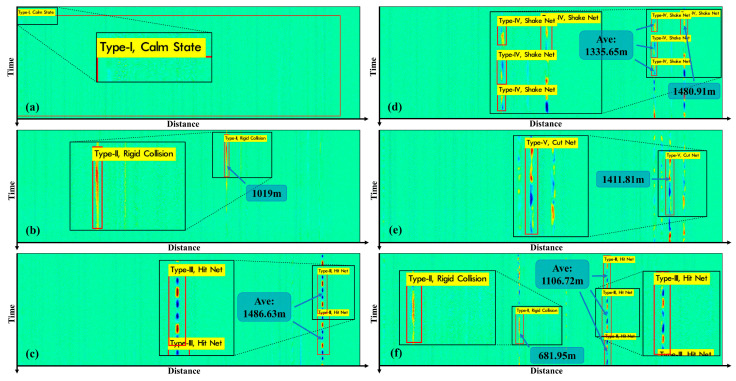
The detection result of “calm state” (**a**), “rigid collision” (**b**), “hit net” (**c**), “shake net” (**d**) and “cut net” (**e**). In (**f**), two types of events are detected at the same time. All black boxes are partial magnifications of the detection results.

**Table 1 sensors-22-01994-t001:** Experiment database: the sample number of each type of event.

Type	I	II	III	IV	V
	Calm State	Rigid Collision	Hit Net	Shake Net	Cut Net
Train set size	560	520	1352	1195	424
Test set size	240	223	580	512	181
Total dataset size	800	743	1932	1707	605

**Table 2 sensors-22-01994-t002:** Performance of algorithms.

Method	Accuracy (%)	Testing Time (sec/img)	Rate (FPS)
Fast R-CNN	95.74%	1.9665	0.5085
Faster R-CNN	97.29%	0.1659	6.0277
YOLO-based	96.14%	0.0438	22.8311

## Data Availability

Not applicable.

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
