# Peer review of "Real-Time Multi-Class Disturbance Detection for Φ-OTDR Based on YOLO Algorithm"

_sensors, 2022, doi:10.3390/s22051994_

Round 1
Reviewer 1 Report
This paper proposes a real-time multi-class disturbance detection algorithm for distributed fiber vibration sensing. The algorithm achieves real-time detection of event location and classification on external intrusions sensed by distributed optical fiber sensing system based on phase-sensitive optical time-domain reflectometry. The work is well done and some quesions need to be clearly explained, as following:
1. How is the exact location of the perturbation calculated in your scheme?
2. In 3.4, real-time is achieved only based on Tp is less than ts? The data processing time should be considered here, and it is recommended to state it in the text.
3. At the end of Section 3.1, there is a typesetting error.
4. There are some grammatical errors in the article, redundant punctuation, spaces, and the case of abbreviations needs to be unified, such as "fps"
5. Figure 6 is not clear enough and needs to be revised.
Reviewer 2 Report
The authors presented a real-time multi-class disturbance detection algorithm based on YOLO for distributed fiber vibration sensing. The manuscript is interesting for the readers in optical fiber sensing. There are some matters to be addressed for publication.
1 There are many versions of the YOLO algorithm. Will using other new versions have better results?
2 In Figure 1 step 5, locating and the location in the description are recommended to be unified.
3 In Figure 4, the action position of "action point" is missing and needs to be filled. Before table 1, there is a typesetting error.
4 A brief description of the event is to be added to Table 1, so that Table 1 and Figure 5 can be more intuitive when combined and compared.
Reviewer 3 Report
The article is devoted to the development of an event detecting system in real-time using an optical fiber, which is a rather urgent task. In general, the work is presented quite well, but there are a number of comments.
First of all, some abbreviations in the abstract (YOLO, Fast-RCNN) are not explained
Is there any estimation of change in algorithm speed after your improvements? The result presented in Table 2 (which by the way is mistakenly labeled as Table 1, line 293) seems kinda random and is not justified.
It's not clear which optical wavelength you use and what is the length of the fiber under test - FUT. By applying an EDFA you are hinting that it is like C-band, but again there is no justification. Maybe 1310 nm can provide more sensitivity due to the large mode diameter?
Regarding EDFA: you claim that the input power is 23 mW with a pulse pump - it is almost the threshold of nonlinear optical effects, have you counted them? Is there a need for such high power? The reader can't estimate it because there is no info about FUT length. Have you combated the noise provided by the input EDFA?
Why do you use 70% of input data as training data? I haven't found any justification.
Line 198, you write about "normal strength and frequency". Are there any real values?
What are the criteria of false alarm, how can the operator prevent it? Regarding the operator: it seems he must know the patterns of each of the 5 events, can you provide some automatic event control?
By the way, there are some technical errors: for example, the manuscript's sidebar is not properly edited, line 28 has two adjacent dots, in lines 190-192 obviously it should be "used" instead of "use", in fig. 7 the matrices represent percents but the scale is ranging from 0 to 1, etc.
